# Safety Risk Assessment of Highway Bridge Construction Based on Cloud Entropy Power Method

**Qingfu Li** [1], **Jianpeng Zhou** [1,*] and **Jinghe Feng** [2]

1 School of Water Conservancy Engineering, Zhengzhou University, Zhengzhou 450001, China
2 Henan Jiaotou Jiaozheng Expressway Co., Ltd., Zhengzhou 450016, China
* Correspondence: 202022222014465@gs.zzu.edu.cn

**Abstract:** (1) In recent years, with China's increasing investment in the transportation industry, the construction of highways and bridges has flourished, bringing great convenience to people's lives. At the same time, there are many uncertain factors in the process of bridge construction, being prone to construction risks. In order to meet the requirements of sustainable development, it is necessary to accurately evaluate the safety risk level of bridge construction. Therefore, it is necessary to establish a new scientific safety risk evaluation system for highway bridge construction. (2) Methods. Based on the relevant standards and specifications, this paper establishes a highway bridge construction safety risk evaluation index system, and then uses the cloud entropy weight method to objectively weight each risk index, using cloud model theory to conduct a risk assessment, and through the cloud model images directly determine the overall risk level of bridge construction, and the level of risk indicators. (3) Results. Applying this method to the construction safety risk assessment of a particular bridge, the overall construction risk level of the bridge is obtained as "level 4", and the risk levels of the four first-level indicators are also all "level 4". (4) Conclusions. The cloud entropy weight method proposed in this paper and the traditional AHP-Extenics method are applied to a bridge construction safety risk evaluation, and the evaluation results obtained are consistent. However, this paper uses the cloud model to improve the entropy weight method in order to calculate the weights, which fully reflects the objectivity of the assignment. The cloud model is used for evaluation, and the risk level of indicators can be determined visually with images.

**Keywords:** cloud entropy weight method; highway bridges; cloud model theory; risk assessment

## 1. Introduction

There are approximately 832,500 highway bridges in China, with a total of 52,256,200 linear meters, and highway bridges have become an important part of the national infrastructure. As the link between highways, bridges are interdependent. Bridge construction is an important and high-risk branch of the transportation construction industry [1]. In particular, the construction of large bridges often faces complex social and natural environments. The environment, coupled with the characteristics of the bridge itself, such as a long construction period, complex structural system, relatively high construction technology, and complex external connections, lead to a series of uncertain factors in the bridge construction process, resulting in construction safety risk, which in turn affects the sustainability of the bridge construction process [2]. During the bridge construction period, risk usually consists of two aspects: one is the size of the possibility of risk occurrence, i.e., risk probability, and the other is the size of the severity of the loss caused by the risk occurrence, also called risk loss. If the risks are not effectively managed and controlled during the bridge construction process, it will not only hinder the progress and cause delays in construction, but even lead to serious economic losses and casualties, which will have a negative impact on the transportation construction industry and society as a whole [3]. To date, several domestic bridge construction accidents have occurred, such as the sudden collapse of the

bridge erection machine for Jiashao Bridge in the prefabricated box girder assembly, and the collapse of Jinshan Bridge in Enshi City, Hubei Province during the pouring process, among others [4]. There are many reasons for these events, involving various aspects such as design, construction, and environmental factors, but in general, the main reasons are the lack of awareness of construction safety risks by construction enterprises, and the lack of risk awareness of a considerable number of construction practitioners [5]. In view of the current research on the safety risk assessment of highway and bridge construction, most studies focus on qualitative or quantitative analysis, which is easily limited to research on the direct risk factors of bridge construction accidents, and lacks in-depth analysis of the nature and mechanism of risk. Bridges are affected by the social and natural environment, construction personnel and management, construction technology, materials and equipment and other factors during construction. These uncertain factors greatly increase the difficulty of construction control. Therefore, it is necessary to establish a more complete and scientific index system for highway and bridge construction safety evaluation; this would reasonably evaluate the safety risk level of highway and bridge construction, reflect the results of risk evaluations in an intuitive way, and make it possible to take corresponding control measures for risk factors in a timely manner, in order to ensure the absolute safety of bridge construction.

At present, scholars at home and abroad have conducted a great deal of research on the safety risk assessment of highway bridge construction, and established many bridge construction safety risk assessment models. In order to accurately and effectively control the construction risk of highway bridges, Yichen Li et al. [6] proposed a method based on network analysis (ANP) and two-dimensional risk assessment methods for cloud models. Seyedmehdi Mortazavi et al. [7] conducted a questionnaire survey on the severity of construction risk indicators among experts and technicians, prioritized risk factors according to the survey results, conducted a quantitative analysis of construction risks on actual bridge projects through Monte Carlo simulation, and determined the impact on bridges of the most critical risk factors for construction safety. Gholamreza Abdollahzadeh [8] used fault tree and event tree analysis methods to evaluate the construction risk of bridge engineering, in order to determine the main causes of fault occurrence and the potential consequences of risk occurrence. Li et al. [9] improved the traditional AHP method and established an entropy-based analytic hierarchy process model, used to evaluate and analyze the risk factors in the bridge construction process. Curra et al. [10] proposed a HYRISK model for bridge scour risk assessment, which was used to determine bridge scour risk probability and consequential loss size, based on the data in the National Bridge Inventory (NBI); the model was successfully used to assess bridge scour in New York State. Yuxin Liu [11] proposed a PC cable-stayed bridge construction based on the fuzzy hierarchical identification method, the F-A-M risk probability assessment method, and the loss equivalent assessment method, using the construction characteristics of the PC cable-stayed bridge. Nieto-Morote [12] proposed a risk assessment method based on the analytic hierarchy process (AHP)-fuzzy set theory using the uncertain factors existing in the bridge construction process. The risk factors were scored and weighted, and then the algorithm was used to deal with the inconsistency in the fuzzy preference relationship in terms of fuzzy judgment. The method was successfully applied to the risk assessment of a bridge project under construction. In order to solve the problems of uncertainty and ambiguity affecting the accuracy of the evaluation results in the safety risk evaluation of bridge construction, Weijun Yang et al. [13] realized qualitative evaluation and quantitative conversion; they proposed a bridge construction safety risk assessment method based on cloud model theory, provided cloud-based evaluation indicators and cloud scale construction methods, and adopted the improved cloud index method to evaluate bridge construction safety risks. Hitoshi Furuta [14] developed a bridge construction risk fuzzy assessment expert system by combining genetic algorithms and neural networks. Considering the difficulties of obtaining expert knowledge on field bridges, the combined method of GA-NN can more accurately evaluate the risk source level. Jianbo Yuan et al. [15] used the network analysis

method to establish a road bridge construction safety risk evaluation model and combined it with the fuzzy comprehensive evaluation method to evaluate the safety risk level of road bridge construction. In the actual assessment process, the existing bridge construction safety risk assessment methods have certain disadvantages, such as an incomplete index system, index weights not matching with the actual engineering, and the assessment results being relatively rough. Therefore, based on the summary of the existing assessment methods, this paper innovatively proposes a highway bridge construction safety risk assessment method based on the cloud entropy weight method.

In this paper, based on the "Analysis of Construction Safety Risk Assessment System and Guidelines for Highway Bridges and Tunnels" [16], the main risk factors during bridge construction are identified from four perspectives: the literature research method, accident causation theory analysis, accident statistical analysis, and construction method analysis. A highway bridge construction safety risk evaluation index system is established, and then the improved cloud entropy weight method is used to objectively assign each risk. The evaluation indexes are then objectively assigned with the traditional weighting method, which fully avoids the subjective arbitrariness of experts' scoring, and makes the calculated weights reflect both the subjective intention of decision makers and the objective properties of data. Finally, by calculating the evaluation cloud of each risk evaluation index, the standard cloud and the comprehensive cloud digital features can be obtained, and then a comprehensive cloud map and standard cloud map can be generated by the cloud forward generator for visual comparison. Then, the cloud forward generator generates a comprehensive cloud map for visual comparison with the standard cloud map, and calculates the closeness $N$ between the comprehensive cloud and the standard cloud to determine the overall risk of highway bridge construction safety and the evaluation level of the primary risk index. In this paper, this method was applied to a bridge construction safety risk assessment; the overall bridge construction safety risk level was four, and the evaluation levels of the four level 1 risk indicators were all also four. In order to verify the scientificity and effectiveness of the method, this paper also combines the AHP-Extenics method to conduct a comparative study on the construction safety risk evaluation of this bridge. The results of the study are consistent with the evaluation results of the cloud entropy weight method used in this paper, which shows that the method is reasonable.

The research flow of this paper is shown in Figure 1.

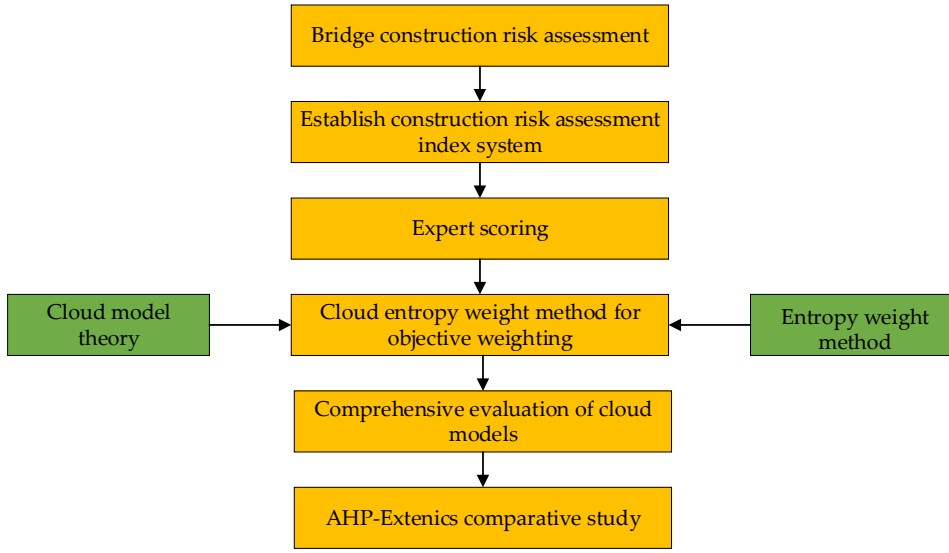

**Figure 1.** Flow chart of the study.

## 2. Selection of Evaluation Indicators

The construction safety risk assessment of highway bridges is a very complex issue. There are many factors affecting the construction safety of bridges, and the interaction

between these factors is very complex. Therefore, the accurate and reasonable identification of the risk factors in the bridge construction process, and the systematic and scientific construction of the highway bridge construction safety risk evaluation index system, will directly affect the accuracy of the evaluation results.

This article refers to "Highway Bridge and Tunnel Construction Safety Risk Assessment System and Guidelines Analysis" to establish a bridge construction safety risk level classification standard, and the bridge risk is divided into five levels, $X = \{X_1, X_2, X_3, X_4, X_5\} = \{$level 1, level 2, level 3, level 4, level 5$\}$; the higher the level, the worse the corresponding bridge safety status, as shown in Table 1.

**Table 1.** Bridge safety state level classification.

| Risk Level | Level 1 | Level 2 | Level 3 | Level 4 | Level 5 |
|---|---|---|---|---|---|
| Security situation | extremely low | Low | medium | high | extremely high |

In this paper, the main risk factors during bridge construction are identified from four perspectives: the literature research method, accident cause theoretical analysis, accident statistical analysis, and construction method analysis [17]. The sources of risk for highway and bridge construction are decomposed into personnel risk, material and equipment risk, construction environment risk, and construction technology risk. Then, through the theoretical analysis of accident causes, accident statistical analysis, and construction method analysis, the relevant risk indicators of personnel risk, material equipment risk, construction environment risk, and construction technology risk are obtained; the important risk factors in these four aspects are determined, and a safety risk assessment index system is established for highway bridge construction, as shown in Figure 2. In addition, economic contract issues, material and equipment transportation, inadequate safety measures, illegal construction operations, the construction monitoring situation, construction site conditions, and other risk factors can adversely affect bridge construction. Due to the relatively small importance of these factors, and the quantification of their weights being complex and error-prone, this paper does not consider them when establishing the bridge construction risk evaluation index system. Sufficient attention still needs to be paid to the bridge construction process, in order to minimize the level of bridge construction risk as much as possible.

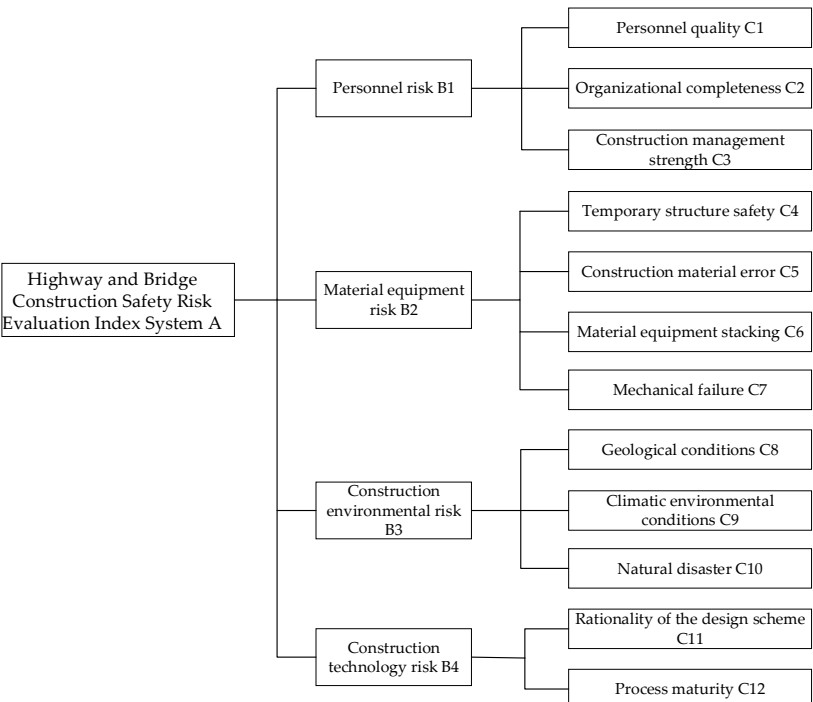

**Figure 2.** Highway bridge construction safety risk evaluation index system.

### 3. Evaluation Method

*3.1. Cloud Model Theory*

3.1.1. Definition of Cloud Model

The cloud model is a transformation model between qualitative concepts and quantitative measures proposed by academician Deyi Li based on traditional fuzzy set theory and probability statistics, which can reveal the intrinsic correlation between the randomness and fuzziness of things themselves. This model is used to study uncertainty and has been applied in river health evaluation, IoT system performance assessment, bridge, and tunnel durability assessment, etc. [17]. At present, the distribution forms of cloud theory that have been developed include triangular clouds, rectangular clouds, trapezoidal clouds, and normal clouds, among which the normal cloud model is widely used due to its unique mathematical properties and universality [18].

Definition of a normal cloud: let $U$ be a theoretical domain, $C$ be a qualitative concept of U with quantitative values $x \in U$, and $x$ be a random realization of $C$ if it satisfies $x \sim N(E_x, E_n{}^2)$, where $E_n{}^2 \sim N(E_n, He^2)$, and $x$ satisfies a certain law for the determinacy $\mu(x)$ of $C$ [19]:

$$\mu(x) = e^{\left[ -\frac{(x-E_x)^2}{2(E_n)^2} \right]} \tag{1}$$

where $E_x$ is the expected value, $E_n$ is the entropy value, and $He$ is the super-entropy value.

3.1.2. Numerical Characteristics of Cloud Model

The cloud model uses the three numerical characteristics of expectation $E_x$, entropy $E_n$, and super-entropy $He$ to comprehensively express a concept of uncertainty [20]:

(1)  Expectation $E_x$: The mathematical expectation value of the distribution of cloud drops in the domain space, i.e., the value of the domain corresponding to the shape of the center area under the affiliated cloud coverage. The cloud drops belonging to this value are located at the highest point of the cloud map, and the affiliation degree is 1. Expectation $E_x$ is the centralized embodiment of the qualitative concept of things; the closer to the expectation point, the more intensive the aggregation of cloud drops, and the higher the recognition of the index.

(2)  Entropy $E_n$: The category of the value of the domain $U$ that the concept can accept, i.e., the measure of the fuzziness of the qualitative concept. Entropy is an important feature used in the cloud model to measure the probability and vagueness of qualitative concepts, reflecting their uncertainty; the larger the entropy value, the more vague the concept is.

(3)  Super-entropy $He$: The uncertainty measure of entropy, i.e., the entropy of entropy, closely related to the randomness and vagueness of the entropy concept, and used to describe the uncertainty of the concept granularity, reflecting the discrete degree of cloud drops.

The schematic diagram of the cloud model digital features is shown in Figure 3.

The cloud model realizes the transformation of qualitative and quantitative uncertainty, and the transformation result is a random number with a stable tendency, which is not a simple superposition of ambiguity and randomness. Different from traditional probability theory and fuzzy theory, the cloud model fully reflects uncertainty in describing things, weakens human subjectivity, and makes the evaluation results more objective, accurate, and rich [21].

3.1.3. Cloud Model Generator

The cloud model generator realizes the interconversion between qualitative concepts and quantitative representations as the core part of cloud model uncertainty reasoning, which is the basis for realizing cloud model control [22]. The cloud model generator takes mainly two forms: the forward cloud model generator and inverse cloud model generator [23]. The forward cloud generator carries out mapping from the qualitative to

quantitative, generating cloud drops from three numerical features of the cloud model ($E_x$, $E_n$, and *He*), and the inverse cloud generator is a model for realizing uncertainty conversion between quantitative values and qualitative language [24], which efficiently converts a certain amount of precise data into concepts, expressed in proper qualitative language values ($E_x$, $E_n$, and *He*) [25]. The operations of the cloud model generator are shown schematically in Figure 4.

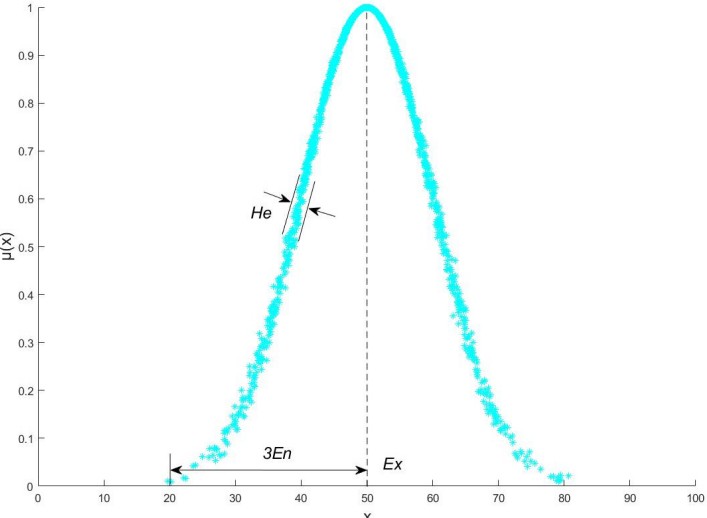

**Figure 3.** Schematic diagram of the digital features of the cloud model.

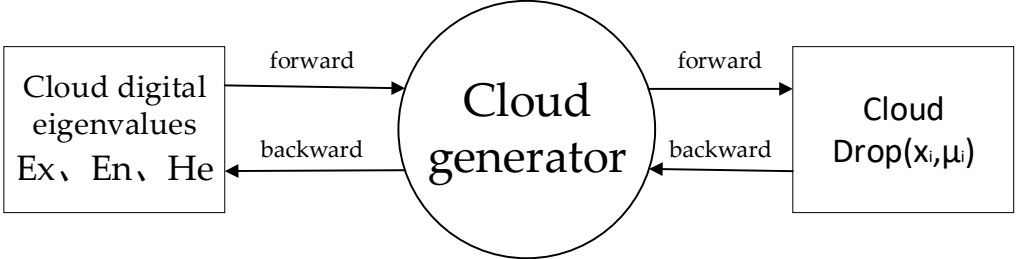

**Figure 4.** Cloud generator operation schematic.

*3.2. Entropy Method*

The entropy weight method is an objective weighting method that determines the weight of decision indicators according to the amount of information contained in each indicator value [26]. Generally, if the entropy corresponding to the index value is smaller, it indicates that the degree of variation of the index value, the impact on the comprehensive evaluation, and the weight value are all greater; otherwise, the weight value is smaller.

There are $m$ evaluation objects, denoted as $M = (M_1, M_2, \cdots, M_m)$, $n$ evaluation indicators, denoted as $N = (N_1, N_2, \cdots, N_n)$, and the value of evaluation object $M_i$ on evaluation indicator $N_j$, denoted as $r_{ij}(i = 1, 2, \cdots, m; j = 1, 2, \cdots, n)$. The original data matrix R is formed by $r_{ij}$, where $r_{ij}$ is the evaluation value of the $i$th object under $j$th evaluation indicators.

The steps for calculating the weights of each evaluation index using the entropy weighting method are as follows [27]:

(1)    Data standardization

The difference in the dimensional units of the evaluation indicators will affect the evaluation results during the calculation process. Therefore, it is necessary to perform dimensionless processing on the original data and obtain the dimensionless matrix $B = \left(b_{ij}\right)_{m \times n}$.

For the benefit-type index with a more favorable value, the dimensionless processing expression is:

$$b_{ij} = \frac{r_{ij} - \min_j(r_{ij})}{\max_j(r_{ij}) - \min_j(r_{ij})} \tag{2}$$

For the cost index, for which the smaller the value, the more favorable it is, the dimensionless processing expression is:

$$b_{ij} = \frac{\max_j(r_{ij}) - r_{ij}}{\max_j(r_{ij}) - \min_j(r_{ij})} \tag{3}$$

where $\max_j(r_{ij})$ is the maximum value of $r_{ij}$ under the $j$th evaluation index, and $\max_j(r_{ij})$ is the minimum value of $r_{ij}$ under the $j$th evaluation index.

(2)  Calculating the proportion of the $i$th evaluation object under the $j$th indicator

$$p_{ij} = \frac{b_{ij}}{\sum\limits_{i=1}^{m} b_{ij}} \tag{4}$$

(3)  Calculating the entropy value of the $j$th indicator

$$e_j = -k \sum_{i=1}^{m} p_{ij} \ln p_{ij} \tag{5}$$

$$K = \frac{1}{\ln m} \tag{6}$$

The greater the difference in a certain index in the evaluation system, the smaller $e_j$; the smaller the difference, the greater $e_j$. If $e_j = 1$, this means that the index $i$ has no influence on the evaluation system at this time.

(4)  Calculating the difference coefficient of the information entropy of the $j$th indicator

$$d_j = 1 - e_j \tag{7}$$

(5)  Determining the weight of each evaluation indicator

$$\gamma_j = \frac{d_j}{\sum\limits_{j=1}^{n} d_j} \tag{8}$$

### 3.3. Cloud Entropy Weight Calculation Model

The cloud model uses three parameters, $E_x$, $E_n$, and $He$, to describe the magnitude, oscillation degree and dispersion degree of the values, with essentially the same idea as the entropy concept, but the data connotation is richer and more consistent with the actual distribution [28]. In this paper, we adopt the cloud model and improve it with reference to the entropy method for the variability of indicators, and obtain the improved model for calculating the indicator weights. The subjective judgments of experts are processed into key representative parameters representing uncertainty by the cloud model, and then the improved calculation model obtains a weight distribution that reflects the importance of risk factors as scientifically as possible, through the organic combination of subjective and objective measures [29].

With $m$ experts and $n$ evaluation metrics, the formula for calculating the numerical eigenvalue of the cloud model for the $j$th evaluation metric is obtained according to the cloud model inverse generator formula, as follows [30]:

$$Ex_j = \overline{x}_j = \frac{1}{m} \sum_{i=1}^{m} x_{ij} \tag{9}$$

$$En_j = \sqrt{\frac{\pi}{2}} \times \frac{1}{m} \sum_{i=1}^{m} |x_{ij} - Ex_j| \tag{10}$$

$$He_j = \sqrt{\frac{1}{m-1} \sum_{i=1}^{m} (x_{ij} - Ex_j)^2 - En_j^2} \tag{11}$$

where $i = 1, 2, \cdots, m; j = 1, 2, \cdots, n$, $Ex_j$, $En_j$, $He_j$ are the expected value, entropy value, and super-entropy value of the $j$th indicator, respectively.

The traditional weight calculation method is used to solve the $j$th evaluation index weight $\beta_j$ as shown in Equation (12).

$$\beta_j = \frac{Ex_j}{\sum\limits_{j=1}^{n} Ex_j} \ (j = 1, 2, \cdots, n) \tag{12}$$

As can be seen from Equation (12), although the traditional weight calculation method is easy to use, it does not make full use of the entropy change in the cloud model and cannot guarantee the comprehensiveness and objectivity of the results. Based on this finding, we propose an improved entropy weighting method to replace this traditional weighting calculation method.

The cloud entropy weighting method is used to solve the $j$th evaluation index weight $\omega_j$, with the following formula [31]:

$$\omega_j = \begin{cases} \dfrac{Ex_j}{\ln(1+En_j)+1} \bullet \dfrac{1}{\sum\limits_{j=1}^{n} \frac{Ex_j}{\ln(1+En_j)+1}} & (En_j \neq 0) \\[4ex] \dfrac{Ex_j}{\sum\limits_{j=1}^{n} Ex_j} & (En_j = 0) \end{cases} \tag{13}$$

### 3.4. Comprehensive Evaluation Model Based on Cloud Theory

The safety risk assessment of bridge construction is a complex system affected by multi-level factors, and the evaluation subject has strong ambiguity and randomness [32]. In this paper, the cloud model theory is used to evaluate the construction safety risk of a bridge. This risk evaluation model is established on the basis of the bridge construction safety risk evaluation index system. The cloud model is used to improve the weight calculation method in order to calculate and evaluate the weight of each risk evaluation index. The standard and evaluation data are converted into cloud models, and the digital features of the standard cloud, evaluation cloud, and comprehensive cloud are obtained. The forward cloud generator is used to generate and compare the comprehensive cloud map and the standard cloud map, and the closeness between the two maps is calculated. The safety risk level of the bridge construction is comprehensively determined.

#### 3.4.1. Standard Cloud $C^m$

The standard cloud $C^m$ is a cloud model generated by the evaluation criteria. In the evaluation of bridge construction safety risk, the evaluation criteria are the criteria for the evaluation indexes to judge the severity level of the evaluation target, which can generally be qualitatively expressed in language, or quantitatively expressed in segmental scores. The normal cloud model is used to calculate the standard cloud, and the steps are as follows:

the evaluation criteria scores are normalized to obtain the effective theoretical domains $U = [X_{\min}, X_{\max}]$, $X_{\min}$ and $X_{\max}$, which indicate the normalized upper and lower limits of the scores belonging to the evaluation levels, respectively, and the corresponding risk levels can be expressed by the cloud numerical characteristic values [33]. The standard cloud $C^m$ is calculated by normalizing the score intervals of the values of different risk levels, $C^m$ as shown in Equation (14).

$$\begin{cases} Ex^m = (X_{\max}^i + X_{\min}^i)/2 \\ En^m = (X_{\max}^i - X_{\min}^i)/2.355 \\ He^m = kEn^m \end{cases} \tag{14}$$

In the formula, $X_{\min}^i$ and $X_{\max}^i$ represent the upper and lower limit values, respectively, of the score interval corresponding to the risk level $i$, and $k$ is the coefficient, which is adjusted according to the fuzzy degree of the concept, being generally taken as 0.1.

### 3.4.2. Evaluation Cloud $C^n$

Evaluation cloud $C^n$ is a cloud model generated from evaluation data. Each quantitative and qualitative evaluation index is scored according to the evaluation judgment criteria, and the data are quantitatively transformed so that the form of the scored values of both indexes is consistent. The raw evaluation data are passed through the cloud inverse generator to generate the numerical characteristics of the evaluation cloud $C^n$ under this criterion; see Equation (15) [34].

$$\begin{cases} Ex^n = \frac{1}{n}\sum\limits_{i=1}^{n} x_i \\ En^n = \frac{1}{n}\sqrt{\frac{\pi}{2}}\sum\limits_{i=1}^{n}|x_i - Ex^n| \\ He^n = \sqrt{S^2 + En^{n2}} \end{cases} \tag{15}$$

In the formula, $x_i$ is the scoring data of the $i$th expert; $S$ is the standard deviation of the scoring data; $Ex^n$ is the expected value of the expert scoring; $En^n$ is the expert scoring entropy value; and $He^n$ is the expert scoring super-entropy value.

### 3.4.3. Integrated Cloud C

Comprehensive cloud C is a parent cloud with deeper meaning obtained by the comprehensive calculation of similar sub-clouds, and its essence is the improvement of concepts [35]. The numerical features of the comprehensive cloud C are obtained by combining the evaluation cloud $C^n$ with the index weight through the fuzzy operator; see Equation (16).

$$\begin{cases} Ex = \sum\limits_{i=1}^{m}(Ex_i \times \omega_i) \\ En = \sqrt{\sum\limits_{i=1}^{m}(En_i^2 \times \omega_i)} \\ He = \sum\limits_{i=1}^{m}(He_i \times \omega_i) \end{cases} \tag{16}$$

In the formula, $Ex_i$, $En_i$, and $He_i$ are the evaluation cloud expectation value, entropy value, and super-entropy value of the first index, respectively.

### 3.4.4. Proximity N

According to the digital features of the calculated comprehensive risk cloud and the standard risk cloud, there is similarity between the generated comprehensive cloud map and the standard cloud map, which may cause a certain visual error in the judgment of the risk evaluation level. Therefore, in order to determine the risk level more accurately, the degree of closeness is used to calculate the degree of closeness between the comprehensive

risk cloud and the standard cloud. The greater the degree of closeness, the closer the evaluation result is to the standard risk level.

$$N = \frac{1}{\sqrt{(Ex - Ex^m)^2}} \tag{17}$$

In the formula, $N$ is the closeness of the risk level, $Ex$ is the expected value of the comprehensive cloud, and $Ex^m$ is the expected value of the standard cloud.

## 4. Project Example Analysis

### 4.1. Project Overview

A highway bridge spans the first-level tributary of the Yangtze River; the width of the river is approximately 300 m, the water level is level 3, and the water level changes greatly with the season. The bridge site is designed on a circular curve with $R$ = 2500 m. The bridge adopts simple support first, and then continuous. The main bridge structure is a 90 m + 170 m + 90 m variable-section prestressed concrete continuous steel box girder, and the approach bridge is a 30 m T-type simply supported girder bridge.

The geological structure of the area where the bridge is located is stable, and there is no fault structure. After investigation, it was found to be mainly composed of sandstone and siltstone. The bridge section contains soil slopes, mainly composed of gravel and clay. It is easily loosened due to the influence of the river water level, and there are certain hidden risks. The bridge site is located in an area of subtropical monsoon climate, and is affected by the corresponding climatic and environmental conditions. The bridge spans rivers and roads and covers a long distance. The terrain is more dangerous, and risk is likely to occur. The bridge construction does not adopt new technology, with the construction technology being relatively mature. The foundation adopts water drilling platform technology, and the upper structure adopts hanging basket construction.

### 4.2. Risk Weight Calculation Based on Cloud Entropy Weight Method

Based on the entropy weight method and the need for the accuracy and validity of cloud model generation [35], 10 experts were invited to score the risk indicators based on two principles: risk probability magnitude and risk loss magnitude. The scoring results are shown in Table 2.

**Table 2.** Risk indicator scoring results.

| Risk Indicator | 1 | 2 | 3 | 4 | 5 | 6 | 7 | 8 | 9 | 10 |
|---|---|---|---|---|---|---|---|---|---|---|
| C1 Personnel quality | 78 | 80 | 65 | 55 | 85 | 43 | 68 | 78 | 72 | 72 |
| C2 Organizational completeness | 67 | 70 | 42 | 80 | 52 | 23 | 60 | 48 | 30 | 86 |
| C3 Construction management strength | 70 | 90 | 65 | 89 | 75 | 60 | 86 | 40 | 60 | 79 |
| C4 Temporary structure safety | 86 | 84 | 56 | 94 | 72 | 67 | 58 | 88 | 72 | 65 |
| C5 Construction material error | 65 | 85 | 50 | 95 | 96 | 73 | 70 | 92 | 90 | 74 |
| C6 Material equipment stacking | 63 | 75 | 40 | 79 | 60 | 55 | 63 | 86 | 84 | 78 |
| C7 Mechanical failure | 30 | 72 | 32 | 80 | 70 | 26 | 56 | 70 | 45 | 75 |
| C8 Geological conditions | 85 | 70 | 78 | 95 | 94 | 68 | 65 | 60 | 58 | 80 |
| C9 Climatic environmental conditions | 75 | 85 | 90 | 97 | 96 | 80 | 80 | 90 | 100 | 81 |
| C10 Natural disasters | 76 | 59 | 35 | 59 | 75 | 56 | 81 | 30 | 82 | 79 |
| C11 Rationality of the design | 75 | 92 | 88 | 65 | 70 | 60 | 65 | 70 | 90 | 95 |
| C12 Process maturity | 60 | 75 | 90 | 80 | 55 | 60 | 52 | 68 | 75 | 95 |

After processing the expert scoring results in the above table through the cloud model inverse generator, the expected score $Ex_j$, the discrete degree $En_j$, and the random degree $He_j$ of each risk factor are obtained according to Equations (9)–(11). Then, according to Equation (12), weight $\beta_j$ can be calculated by the traditional weight calculation method, and the improved weight $\omega_j$ obtained by using the improved entropy weight Equation (13). The results are shown in Table 3.

**Table 3.** Objective weighting of risk indicators.

| Risk Indicator | $Ex_j$ | $Ex_n$ | $He$ | $\omega_j$ | $\beta_j$ |
|---|---|---|---|---|---|
| C1 Personnel quality | 69.600 | 10.152 | 1.791 | 0.091 | 0.082 |
| C2 Organizational completeness | 55.800 | 23.011 | 6.243 | 0.059 | 0.066 |
| C3 Construction management strength | 71.400 | 15.892 | 1.734 | 0.083 | 0.084 |
| C4 Temporary structure safety | 74.200 | 14.438 | 5.015 | 0.088 | 0.088 |
| C5 Construction material error | 79.000 | 15.629 | 4.246 | 0.092 | 0.093 |
| C6 Material equipment stacking | 68.300 | 13.573 | 3.008 | 0.083 | 0.081 |
| C7 Mechanical failure | 55.600 | 24.878 | 4.212 | 0.058 | 0.066 |
| C8 Geological conditions | 75.300 | 15.428 | 5.133 | 0.088 | 0.089 |
| C9 Climatic environmental conditions | 87.400 | 10.202 | 2.600 | 0.114 | 0.103 |
| C10 Natural disasters | 63.200 | 21.031 | 6.530 | 0.069 | 0.075 |
| C11 Rationality of the design | 77.000 | 14.288 | 5.956 | 0.092 | 0.091 |
| C12 Process maturity | 71.000 | 15.040 | 3.617 | 0.084 | 0.084 |

The analysis of Table 3 shows that the expected value of each risk index has a good distribution, and the interval difference is more obvious; the entropy value and the super-entropy value represent the dispersion degree of the expert opinions. The risk weight value obtained by the weight calculation method is generally 6% to 9%, and the weight value distribution is relatively even. Such a result is not conducive to the distinction of primary and secondary weights, and is not conducive to the judgment of factor importance.

The improved weights of each risk index are sorted from small to large, as shown in Figure 5.

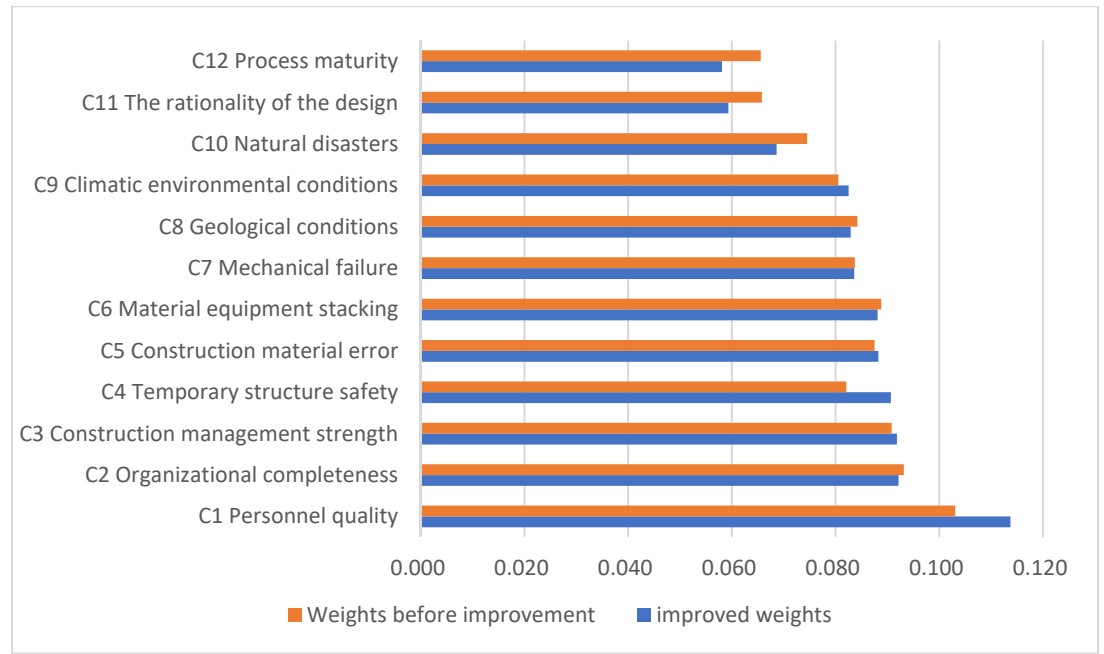

**Figure 5.** Risk indicator weighting chart.

As can be seen from the above figure:

(1) The ranking of the importance of indicators has not changed much after the improvement, and basically maintains the trend of the original expected value ranking, which fully reflects the opinions of experts.

(2) The weight distribution is readjusted by the difference in the scoring results, which shows that the improved model makes fuller use of the scoring data results and more effectively reflects the objective scoring situation.

(3)     After the improvement, the difference in the weight distribution becomes larger, and the weight after the improvement is generally between 5% and 11%. The weight range between the top and bottom indicators nearly doubles, which can avoid the problem of averaging weight values.

In summary, the traditional weight calculation method using Equation (12) only uses the expected value $E_x$ in the cloud model for calculation, without making full use of the variation in entropy in the cloud model, and cannot guarantee the objectivity of the results. If the expected values $E_x$ of all indicators are not very different, the calculated weight value will tend to be averaged and cannot objectively reflect the actual situation. However, in fact, the entropy $E_n$ of each indicator varies greatly. Based on this finding, this paper proposes an improved entropy weighting method of the cloud model shown in Equation (13) to replace this traditional weighting method, which can effectively avoid the problem of averaging the weights of evaluation indicators.

### 4.3. Risk Assessment Analysis

#### 4.3.1. Determination of Standard Cloud

In this paper, the cloud model is used to evaluate the safety risk of bridge construction. In order to facilitate the expert judgment and scoring, a unified judgment criterion is used. This paper refers to the "Guidelines for Safety Risk Assessment of Highway Bridge and Tunnel Engineering Construction", and combines the construction site safety specifications and expert opinions. The construction safety risk is divided into five levels according to the possibility and severity of the risk, and is described in qualitative language: extremely low, low, moderate, high, and extremely high.

From the risk level classification standard in Table 1, the standard cloud digital characteristics are calculated by Equation (14), and the standard risk cloud map is generated by the forward cloud generator. The digital characteristics are shown in Table 4, and the standard cloud map is shown in Figure 6.

**Table 4.** Risk evaluation criteria and standard cloud digital features.

| Risk Level | Level 1 | Level 2 | Level 3 | Level 4 | Level 5 |
|---|---|---|---|---|---|
| Security situation | very low | low | middle | high | very high |
| Risk consequences | Ignorable | small | general | serious | essential |
| Score | $(0-20)$ | $(21-40)$ | $(41-60)$ | $(61-80)$ | $(81-100)$ |
| Cloud digital features | $(10, 8.49, 0.85)$ | $(30, 8.49, 0.85)$ | $(50, 8.49, 0.85)$ | $(70, 8.49, 0.85)$ | $(90, 8.49, 0.85)$ |

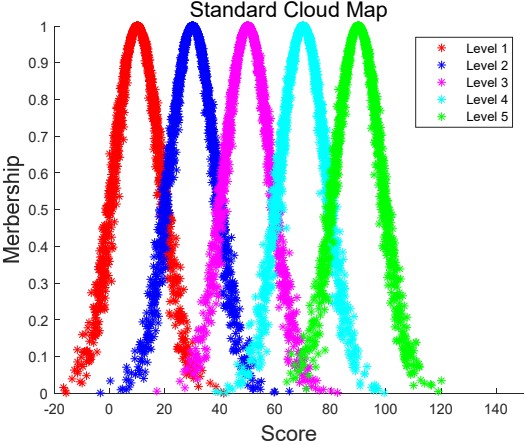

**Figure 6.** Standard cloud map.

#### 4.3.2. Determination of Safety Risk Assessment Level in Bridge Construction

In this paper, 10 experts were invited to score the risk indicators. The scoring results are shown in Table 2. According to Equation (15) and the weight determined by the

abovementioned cloud entropy weight method, to determine the evaluation cloud digital features, the comprehensive cloud digital features are obtained using Equation (16), and the calculation results are shown in Table 5.

**Table 5.** Weights and cloud numerical feature results.

| First-Level Indicator | Weights | Cloud Digital Features | Secondary Indicators | Weights | Cloud Digital Features |
|---|---|---|---|---|---|
| B1 | 0.233 | (66.725, 16.277, 2.905) | C1 | 0.091 | (69.60, 10.15, 1.79) |
| | | | C2 | 0.059 | (55.80, 23.01, 6.24) |
| | | | C3 | 0.083 | (71.40, 15.89, 1.73) |
| | | | C4 | 0.088 | (74.20, 14.44, 5.02) |
| B2 | 0.321 | (70.695, 16.935, 4.133) | C5 | 0.092 | (79.00, 15.63, 4.25) |
| | | | C6 | 0.083 | (68.30, 13.57, 3.01) |
| | | | C7 | 0.058 | (55.60, 24.88, 4.21) |
| | | | C8 | 0.088 | (75.30, 15.43, 5.13) |
| B3 | 0.271 | (77.317, 15.282, 4.423) | C9 | 0.114 | (87.40, 10.20, 2.60) |
| | | | C10 | 0.069 | (63.20, 21.03, 6.53) |
| B4 | 0.176 | (74.141, 14.651, 4.841) | C11 | 0.092 | (77.00, 14.29, 5.96) |
| | | | C12 | 0.084 | (71.00, 15.04, 3.62) |

From the weights of the first-level indicators and the comprehensive cloud digital features, the target layer comprehensive cloud digital features can be further calculated. From Equation (16), the target layer comprehensive cloud digital features are (72.166, 15.958, 4.050).

The digital features of the first-level index integrated cloud and the target layer integrated cloud calculated above are used to generate a comprehensive cloud map through the cloud forward generator, which is compared with the standard cloud map to determine the risk level of the target layer and the first-level index. The comparison chart with the standard cloud is shown in Figures 7–11.

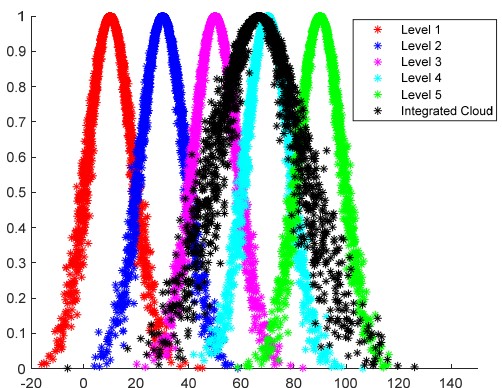

**Figure 7.** People risk integrated cloud versus standard cloud.

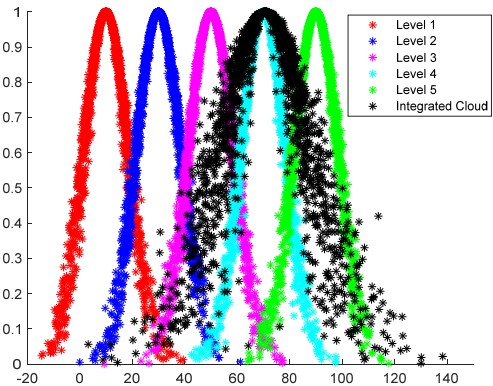

**Figure 8.** Material and equipment risk integrated cloud vs. standard cloud.

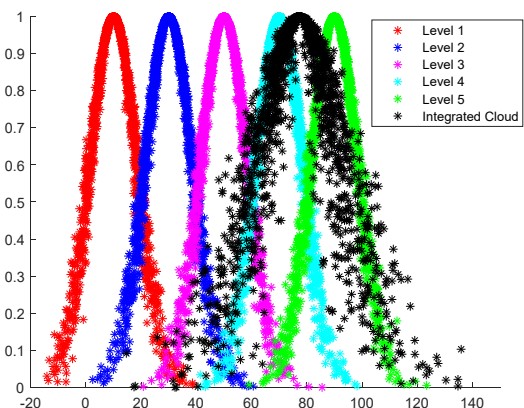

**Figure 9.** Construction environmental risk integrated cloud vs. standard cloud.

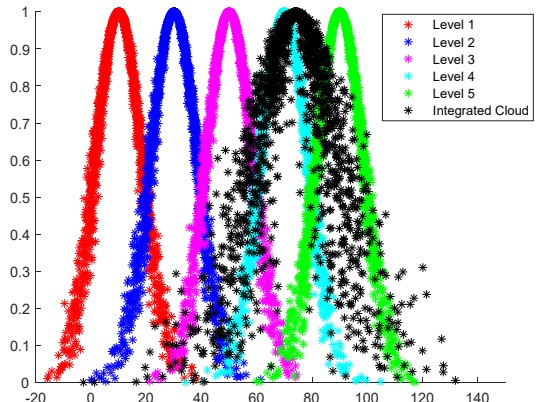

**Figure 10.** Construction technology risk integrated cloud vs. standard cloud.

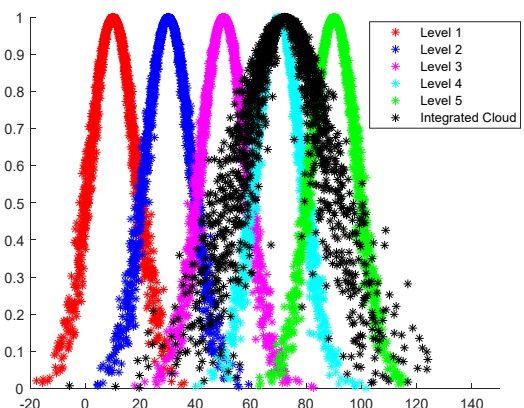

**Figure 11.** Comparison of bridge construction safety risk integrated cloud and evaluation cloud.

Due to the similarity between the comprehensive risk cloud and the standard risk cloud, and that fact that one-dimensional graphics can cause visual errors, only the risk level can be determined initially. Therefore, this paper introduces the closeness degree to accurately determine the construction risk level of the bridge; for the calculation result of the closeness degree, see Table 6.

From Figures 7–10 it can be seen that the risk level of personnel risk B1 is between level 3 and level 4, being closer to level 4, and the degree of closeness between the B1 comprehensive cloud and level 4 standard cloud is the largest, with a value of 0.305. The risk level of material equipment risk B2 is close to level 4, and the closeness of the B2 comprehensive cloud and level 4 standard cloud is the largest, with a value of 1.440. The risk level of construction environment risk B3 is between level 4 and level 5, being closer

to level 4, and the degree of closeness between the B3 comprehensive cloud and level 4 standard cloud is the largest, with a value of 0.137. The risk level of construction technology risk B4 is between level 4 and level 5, being closer to level 4, and the B4 comprehensive cloud is closer to level 4. The closeness of the level 4 standard cloud is the largest, with a value of 0.242. Therefore, the risk levels of the four first-level indicators are all four, and the risk levels are ranked as follows: construction environment risk B3 > construction technology risk B4 > material and equipment risk B2 > personnel risk B1.

**Table 6.** Proximity calculation results.

| Risk Level | Risk Indicator | | | | |
|---|---|---|---|---|---|
| | B1 | B2 | B3 | B4 | A |
| Level 1 | 0.018 | 0.016 | 0.015 | 0.016 | 0.016 |
| Level 2 | 0.027 | 0.025 | 0.021 | 0.023 | 0.024 |
| Level 3 | 0.060 | 0.048 | 0.037 | 0.041 | 0.045 |
| Level 4 | 0.305 | 1.440 | 0.137 | 0.242 | 0.462 |
| Level 5 | 0.043 | 0.052 | 0.079 | 0.063 | 0.056 |

According to the comparison chart of the comprehensive cloud and the evaluation cloud of bridge construction safety risk in Figure 11, the overall construction safety risk level of the bridge is between level 4 and level 5, which is very close to the center expectation of level 4, between 61 and 80. The bridge construction safety risk comprehensive cloud has the highest similarity with the four-level standard cloud, so it can be determined that the bridge construction safety risk level is four. Therefore, the formulation of the corresponding control measures during construction shall focus on the abovementioned higher-level safety risk factors.

In conclusion, the entropy weight method in conjunction with cloud model theory has been well tested in the actual safety risk assessment of bridge construction, which reflects the accuracy and applicability of the method, providing a new method for highway bridge construction safety risk assessment.

### 4.4. Comparative Study

In order to illustrate the rationality and scientificity of the method, this paper uses the AHP-Extenics method to perform a comparative study.

### 4.4.1. Analytic Hierarchy Process (AHP)

The AHP method is a subjective weighting analysis method that solves problems in which it is difficult to accurately recognize the relative importance of multiple targets. By measuring the relative importance of each target, the weighting can be reasonably quantified, and the weight value can be used to effectively compare the relative importance of each target [36]. The calculation steps of the AHP to determine the weight of the evaluation index are as follows:

Constructing a Judgment Matrix

After stratifying each evaluation index, the relative importance of each index at the same level is compared, and assigned by the method of pairwise comparison to determine the relative importance of each index at the lower level compared with the index at the upper level, so as to obtain the weight of each evaluation index. If a certain layer has *n* evaluation indicators, the judgment matrix can be constructed as shown in Equation (19) [37].

$$A = (a_{ij})_{n \times n} = \begin{bmatrix} a_{11} & \cdots & a_{1n} \\ \cdots & \cdots & \cdots \\ a_{n1} & \cdots & a_{nn} \end{bmatrix} \tag{18}$$

where $i = 1, 2, \cdots, n$; $j = 1, 2, \cdots, n$; $a_{ij}$ indicates the judgment result of the comparative importance of two evaluation indicators $i$ and $j$ in criterion A. In this paper, the expert scoring method is used to determine the original data based on the 1–9 scale method [38] to compare the evaluation indicators, which can quantify the relative importance between the two factors. The values and importance relationships are shown in Table 7, values 2, 4, 6, and 8 indicate that the relative factor importance is between two adjacent levels.

**Table 7.** Relationship between numerical value and importance.

| Numerical Value | Importance |
|:---:|:---:|
| 1 | Equal importance |
| 3 | Moderate importance of one over another |
| 5 | Essential or strong importance |
| 7 | Very strong importance |
| 9 | Extreme importance |

Calculating the Weight of the Evaluation Index

The eigenvector $\beta$ corresponding to the maximum eigenvalue $\lambda_{\max}$ is calculated according to the judgment matrix $A$ obtained by the pairwise comparison of the evaluation indicators; then, the calculation problem of the evaluation index weight is transformed into the problem of solving the eigenvector $\beta$ of the judgment matrix $A$. The equation is as follows:

$$A\alpha = \lambda_{\max}\alpha \tag{19}$$

In this paper, the square root method is used to calculate the weight, and the specific calculation steps are as follows [39]:

Step 1: Each row of the judgment matrix $A$ is multiplied to obtain $u_i$; see Equation (20).

$$u_i = \prod_{j=1}^{n} a_{ij}, i = 1, 2, \cdots n \tag{20}$$

Step 2: $u_i$ is squared $n$ times separately to obtain $u_i'$; see Equation (21).

$$u_i' = \sqrt[n]{u_i} \tag{21}$$

Step 3: $u_i'$ is regularized to obtain the feature vector $\alpha_i$; see Equation (22).

$$\alpha_i = \frac{u_i'}{\sum\limits_{i=1}^{n} u_i'} \tag{22}$$

Step 4: The maximum characteristic root $\lambda_{\max}$ of the judgment matrix $A$ is calculated; see Equation (23).

$$\lambda_{\max} = \sum_{i=1}^{n} \frac{(A\alpha)_i}{n\alpha_i} \tag{23}$$

Consistency Test

In order to verify the rationality of the above weight calculation, it is necessary to perform a consistency check on the judgment matrix $A$; the specific steps are as follows [40]:

Step 1: The consistency index $C.I.$ is calculated; see Equation (24).

$$C.I. = \frac{\lambda_{\max} - n}{n - 1} \tag{24}$$

where $n$ is the order of judgment matrix $A$; $\lambda_{\max}$ is the maximum characteristic root of judgment matrix $A$.

Step 2: The consistency ratio *C.R.* is calculated; see Equation (25).

$$C.R. = \frac{C.I.}{R.I.} \tag{25}$$

where *R.I.* is the average random consistency index, which can be determined by checking Table 8 [41].

**Table 8.** Average random consistency index, *R.I.*

| *n* | 1 | 2 | 3 | 4 | 5 | 6 | 7 | 8 | 9 | 10 |
|---|---|---|---|---|---|---|---|---|---|---|
| R.I. | 0 | 0 | 0.52 | 0.89 | 1.12 | 1.26 | 1.36 | 1.41 | 1.46 | 1.49 |

According to the consistency test standard, when the consistency ratio C.R. < 0.10, it means that the constructed judgment matrix satisfies the consistency test; otherwise, it is necessary for the experts to re-score and construct a new judgment matrix until the test is satisfied.

### 4.4.2. Extenics Theory

Extenics theory is a new discipline founded by Cai Wen and other scholars from the Guangdong University of Technology. It uses formal models to study the possibility of things expanding and the laws and methods of innovation, and is widely used in multi-objectives. The comprehensive judgment problem is mainly divided into matter-element theory, extension set theory, and extension logic. Among them, matter-element theory has great advantages in solving the uncertainty and ambiguity of things. Matter-element theory mainly includes the following three parts: the determination of the classical domain and section domain, the determination of the matter-element to be evaluated, and the calculation of the correlation degree of the evaluation index [42].

Let the risk assessment research of highway bridge construction be matter-element $R$, under which there are $m$ evaluation indexes, and at the same time, each evaluation index $u_i$ is divided into $n$ evaluation levels; then, the evaluation index set $U = \{u_1, u_2, \cdots, u_m\}$ and the evaluation level set $V = \{v_1, v_2, \cdots, v_n\}$.

(1) The object element classical domain is denoted as:

$$R_j = (V_j, u_i, x_{ij}) = \begin{bmatrix} V_j, & u_1, & x_{1j} \\ & u_2 & x_{2j} \\ & \vdots & \vdots \\ & u_m & x_{mj} \end{bmatrix} = \begin{bmatrix} V_j, & u_1 & \langle a_{1j}, b_{1j} \rangle \\ & u_2 & \langle a_{2j}, b_{2j} \rangle \\ & \vdots & \vdots \\ & u_m & \langle a_{mj}, b_{mj} \rangle \end{bmatrix} \tag{26}$$

where $V_j$—the $j$th ($j = 1, 2, \cdots, n$) level of the evaluation object; $u_i$—the $i$th ($i = 1, 2, \cdots, m$) evaluation index of the evaluation object; $x_{ij}$—the value of the corresponding $u_i$ index when the evaluation object belongs to the $j$th level; $\langle a_{ij}, b_{ij} \rangle$—the value range when the evaluation object $i$ belongs to the $j$th evaluation level, that is, the classical domain; $a_{ij}$ is the lower limit of the value of the $i$th index $u_i$; and $b_{ij}$ is the upper limit of the value of the $i$th index $u_i$.

The nodal domain of the object element is expressed as:

$$R_P = [P, u_i, x_{iP}] = \begin{bmatrix} P, & u_1, & x_{1P} \\ & u_2 & x_{2P} \\ & \vdots & \vdots \\ & u_m & x_{mP} \end{bmatrix} = \begin{bmatrix} V_j, & u_1, & \langle a_{1P}, b_{1P} \rangle \\ & u_2 & \langle a_{2P}, b_{2P} \rangle \\ & \vdots & \vdots \\ & u_m & \langle a_{mP}, b_{mP} \rangle \end{bmatrix} \tag{27}$$

where $P$—the whole of the evaluation level; $x_{ip} = \langle a_{ip}, b_{ip} \rangle$—the range of all values of evaluation index $u_i$, i.e., the section field; $a_{ip}$ is the minimum value of the lower limit of the

ith evaluation index $u_i$ in all evaluation levels; and $b_{ip}$ is the maximum value of the upper limit of the *i*th index $u_i$ in all evaluation levels [43].

(2) Determination of the elements to be evaluated

$$R_W = \begin{bmatrix} W & u_1 & x_1 \\ & u_2 & x_2 \\ & \vdots & \vdots \\ & u_n & x_n \end{bmatrix} \tag{28}$$

where *W*—an object to be evaluated; and $x_i$—*W* for the evaluation of the value range of index $u_i$, the specific index data of the object to be evaluated.

(3) Calculation of the correlation of the evaluation index

The correlation degree of the construction safety risk level *j* of the highway and bridge to be evaluated is as follows:

$$\rho(x_i, x_{ij}) = \left| x_i - \frac{a_{ij} + b_{ij}}{2} \right| - \frac{b_{ij} - a_{ij}}{2} \tag{29}$$

$$\rho(x_i, x_{ip}) = \left| x_i - \frac{a_{ip} + b_{ip}}{2} \right| - \frac{b_{ip} - a_{ip}}{2} \tag{30}$$

$$K_j(x_i) = \frac{\rho(x_i, x_{ij})}{\left[\rho(x_i, x_{ip}) - \rho(x_i, x_{ij})\right]} \tag{31}$$

where $K_j(x_i)$—the correlation degree of the object to be evaluated when the evaluation level is *j*.

(4) Determining the safety risk assessment level of highway bridge construction

Combined with the weight coefficients of each index, and the calculated correlation function value synthesized to obtain the evaluation object's level of correlation shown in Equation (32), if $K_{\max} = K_j$, then the evaluation object belongs to level *j*.

$$K_j(W) = \sum_{i=1}^{n} \alpha_i K_j(x_i) \tag{32}$$

where $\alpha_i$ is the weight value of evaluation index $u_i$; the larger $\alpha_i$ is, the greater the degree of influence of the evaluation index on the evaluation object.

### 4.4.3. Risk Indicator Weight Calculation

According to the highway and bridge construction safety risk evaluation index system presented in Figure 1, the 1–9 scale method is used to construct the judgment matrix of each level index, and the weight of the highway bridge construction safety risk evaluation index is calculated. The judgment matrices and their weight assignments of the target layer A–B and the criterion layers B1–C, B2–C, and B3–C are listed in Tables 9–13, respectively.

**Table 9.** A–B judgment matrix and weight assignment.

| A | B1 | B2 | B3 | B4 | Wi | Consistency Check |
|---|----|----|----|----|-----|-------------------|
| B1 | 1 | 1 | 0.5 | 1 | 0.195 | $\lambda$max: 4.0458 |
| B2 | 1 | 1 | 0.5 | 2 | 0.231 | |
| B3 | 2 | 2 | 1 | 3 | 0.426 | C.R. = 0.0172 < 0.1 |
| B4 | 1 | 0.5 | 0.333 | 1 | 0.148 | |

**Table 10.** B1–C judgment matrix and weight assignment.

| B1 | C1 | C2 | C3 | Wi | Consistency Check |
|----|----|----|----|----|-------------------|
| C1 | 1 | 3 | 2 | 0.528 | $\lambda$max: 3.0536 |
| C2 | 0.333 | 1 | 0.333 | 0.140 | |
| C3 | 0.5 | 3 | 1 | 0.333 | C.R. = 0.0516 < 0.1 |

**Table 11.** B2–C judgment matrix and weight assignment.

| B2 | C4 | C5 | C6 | C7 | Wi | Consistency Check |
|----|----|----|----|----|----|-------------------|
| C4 | 1 | 1 | 1 | 2 | 0.272 | $\lambda$max: 4.0813 |
| C5 | 1 | 1 | 2 | 3 | 0.362 | |
| C6 | 1 | 0.5 | 1 | 3 | 0.255 | |
| C7 | 0.5 | 0.333 | 0.333 | 1 | 0.111 | C.R. = 0.0304 < 0.1 |

**Table 12.** B3–C judgment matrix and weight assignment.

| B3 | C8 | C9 | C10 | Wi | Consistency Check |
|----|----|----|-----|----|-------------------|
| C8 | 1 | 2 | 2 | 0.493 | $\lambda$max: 3.0536 |
| C9 | 0.5 | 1 | 0.5 | 0.196 | |
| C10 | 0.5 | 2 | 1 | 0.311 | C.R. = 0.0516 < 0.1 |

**Table 13.** B4–C judgment matrix and weight assignment.

| B4 | C11 | C12 | Wi | Consistency Check |
|----|-----|-----|----|-------------------|
| C11 | 1 | 2 | 0.667 | $\lambda$max: 2.0000 |
| C12 | 0.5 | 1 | 0.333 | C.R. = 0 < 0.1 |

From the calculation results in Tables 8–12, it can be seen that in the safety risk assessment analysis of highway bridge construction, the index weight of the target layer A–B is $W_{A-B} = (0.195, 0.231, 0.426, 0.148)^T$, and the index weight of the criterion layer B–C is $W_{B-C}$ = (0.195, 0.231, 0.426, 0.148, 0.528, 0.140, 0.333, 0.272, 0.362, 0.255, 0.111, 0.493, 0.196, 0.311, 0.667, 0.333)T. The consistency test indexes of each layer judgment matrix C.R. < 0.1 all meet the consistency requirements. On this basis, the comprehensive weight of each risk index is calculated and then sorted, and the overall ranking of the road and bridge construction safety risk assessment index levels can be obtained. The results are listed in Table 14.

**Table 14.** Total hierarchical ranking of highway bridge construction safety risk evaluation indexes.

| Target Layer | First-Level Indicator | Primary Weight | Secondary Indicator | Secondary Weight | Comprehensive Weight | Rank |
|--------------|----------------------|----------------|---------------------|------------------|---------------------|------|
| A | B1 | 0.195 | C1 | 0.528 | 0.103 | 5 |
| | | | C2 | 0.140 | 0.027 | 12 |
| | | | C3 | 0.333 | 0.065 | 8 |
| | B2 | 0.231 | C4 | 0.272 | 0.116 | 2 |
| | | | C5 | 0.362 | 0.154 | 1 |
| | | | C6 | 0.255 | 0.109 | 4 |
| | | | C7 | 0.111 | 0.047 | 10 |
| | B3 | 0.426 | C8 | 0.493 | 0.072 | 7 |
| | | | C9 | 0.196 | 0.114 | 3 |
| | | | C10 | 0.311 | 0.045 | 11 |
| | B4 | 0.148 | C11 | 0.667 | 0.099 | 6 |
| | | | C12 | 0.333 | 0.050 | 9 |

It can be seen from Table 13 that among the safety risk evaluation indicators of highway bridge construction, the type of risk that has the greatest impact on highway bridge construction safety is construction material error C5, followed by temporary structure safety C4, climatic and environmental conditions C9, and material and equipment stacking C6.

#### 4.4.4. Comprehensive Evaluation of AHP-Extenics

In this paper, referring to the "Analysis of Safety Risk Assessment System and Guidelines for Highway Bridge and Tunnel Engineering Construction", combined with construction site safety regulations and expert opinions, the safety risk of bridge construction is divided into five levels: level 1 (0–20), level 2 (21–40), level 3 (41–60), level 4 (61–80) and level 5 (81–100). From this, it is determined that the construction safety risk level U= {$X_1$, $X_2$, $X_3$, $X_4$, $X_5$} = {level 1, level 2, level 3, level 4, level 5}.

This paper invited 10 experts to score the risk indicators. The scoring results are shown in Table 2. In order to avoid the influence of the subjectivity of experts' scoring on the evaluation results, this paper takes the expected value of each expert's score for a certain index to calculate the relevance of the index. The classic domain and section domain in the safety risk evaluation index of highway bridge construction are shown in Table 15, and the expected value of the expert scoring of each evaluation index is shown in Table 16.

**Table 15.** Classical and sectional domains in the construction safety risk evaluation index.

| Evaluation Indicators | Classic Domain | | | | | Sectional Domain |
|---|---|---|---|---|---|---|
| | $X_1$ | $X_2$ | $X_3$ | $X_4$ | $X_5$ | |
| C1~C12 | [0, 20) | [20, 40) | [40, 60) | [60, 80) | [80, 100) | [0, 100) |

**Table 16.** Expert scoring expectations for each evaluation index.

| Evaluation Indicators | C1 | C2 | C3 | C4 | C5 | C6 | C7 | C8 | C9 | C10 | C11 | C12 |
|---|---|---|---|---|---|---|---|---|---|---|---|---|
| Scoring expectations | 69.6 | 55.8 | 71.4 | 74.2 | 79 | 68.3 | 55.6 | 75.3 | 87.4 | 63.2 | 77 | 71 |

From Equations (30)–(32), the correlation degree of the safety risk evaluation index for the construction of highway bridges to be evaluated can be obtained. The calculation results are as follows:

$$K_j(x_i) = \begin{bmatrix} -0.620 & -0.493 & -0.240 & 0.462 & -0.255 \\ -0.448 & -0.263 & 0.105 & -0.087 & -0.354 \\ -0.643 & -0.523 & -0.285 & 0.430 & -0.231 \\ -0.678 & -0.570 & -0.355 & 0.290 & -0.184 \\ -0.738 & -0.650 & -0.475 & 0.050 & -0.045 \\ -0.604 & -0.472 & -0.208 & 0.355 & -0.270 \\ -0.445 & -0.260 & 0.110 & -0.090 & -0.355 \\ -0.691 & -0.588 & -0.383 & 0.235 & -0.160 \\ -0.843 & -0.790 & -0.685 & -0.370 & 1.423 \\ -0.975 & -0.955 & -0.744 & -1.524 & -0.939 \\ -0.713 & -0.617 & -0.425 & 0.150 & -0.115 \\ -0.638 & -0.517 & -0.275 & 0.450 & -0.237 \end{bmatrix}$$

The subjective weight values of each secondary index calculated by the AHP are $W_{B1–C}$ = (0.528,0.140,0.333), $W_{B2–C}$ = (0.272,0.362,0.255,0.111), $W_{B3–C}$ = (0.493, 0.196, 0.311), and $W_{B4–C}$ = (0.667, 0.333). According to Equation (32), the comprehensive correlation degree of the criterion layer for the evaluation level j of the highway bridge construction safety risk can be calculated as:

$$K_j(B_1) = (-0.603, -0.471, -0.207, 0.374, -0.261)$$
$$K_j(B_2) = (-0.655, -0.539, -0.309, 0.177, -0.174)$$
$$K_j(B_3) = (-0.809, -0.742, -0.554, -0.043, -0.092)$$
$$K_j(B_4) = (-0.688, -0.583, -0.375, 0.250, -0.156)$$

The weight value of the first-level index obtained by the above AHP is $W_{A-B}$ = (0.195, 0.231, 0.426, 0.148). Similarly, according to Equation (32), the safety risk of the highway and bridge construction can be calculated to obtain the target of the evaluation level j. The comprehensive correlation degree of the layer is:

$$K_j(A) = (-0.715, -0.619, -0.403, -0.032, -0.153)$$

According to the principle of maximum membership, the risk levels of the four first-level indicators of the criterion layer are personnel risk B1 (level 4), material and equipment risk B2 (level 4), construction environment risk B3 (level 4), and construction technology risk B4 (level 4); the overall construction safety risk level of the bridge is also level 4.

In this study, two different methods were used to evaluate the safety risk of bridge construction. The evaluation results are consistent, and the evaluation grades are all four. In the controlled study, the traditional AHP method was used to calculate the weights. Due to the variable level of the experts, it is difficult to ensure the scientificity and accuracy of the weights. In this paper, the cloud model was used to improve the entropy weight method to objectively weight each risk evaluation index, and the subjective judgment of experts was processed into the key representative parameters of uncertainty through the cloud model. The subjective and objective weight distribution, through the organic combination of subjective and objective measures, reflects the importance of each risk factor as scientifically as possible. The cloud map was compared with the standard cloud map, and the closeness of the comprehensive cloud and the standard cloud was calculated to comprehensively determine the risk level of the target layer and the first-level indicators. The evaluation results obtained are consistent with the results determined by AHP-Extenics. Through comparative research, the rationality of the method proposed in this paper is highlighted.

## 5. Conclusions

Highway bridge construction risk evaluation is an important part of bridge construction, and it is therefore of great significance to propose a scientific and effective highway bridge construction risk evaluation method for the construction safety and normal use of bridges. In this paper, with reference to the literature, we use the cloud entropy power method to evaluate the construction risk of a bridge, and the main conclusions are as follows:

(1) Referring to the relevant standards and specifications, this paper decomposes the highway bridge construction risk sources into four first-level risk indicators—personnel risk, material and equipment risk, construction environment risk, and construction technology risk—and selects personnel quality, organizational completeness, construction management strength, temporary structure safety, construction material error, material and equipment stacking, machinery failure, geological conditions, climatic and environmental conditions, and natural disasters. Then, the cloud entropy weight method was used to objectively assign weight to each risk indicator, and was compared to the traditional weighting method.

(2) To determine the evaluation level of highway bridge construction risk factors, the evaluation criteria and evaluation data were transformed into a cloud model, and the forward cloud generator was used to generate and compare a comprehensive cloud map and a standard cloud map. Then, the closeness N between the two maps was calculated to comprehensively determine the bridge construction safety risk levels as follows: personnel risk B1 (level 4), material and equipment risk B2 (level 4), construction environment risk B3 (level 4), construction technology risk B4 (level 4), and overall construction safety risk (level 4). The method proposed here was also combined with the AHP-Extenics method to perform a comparative study, and the assessment results are consistent, which proves that the cloud entropy weight method can be used in the evaluation of highway bridge construction risks with certain scientific validity.

(3) This paper evaluates the construction risk of highway bridges. Since there are many risk factors affecting bridge construction safety, this paper only selects four primary risk indicators and twelve secondary risk indicators that are more common and have a greater impact on construction safety. There are shortcomings in many aspects, such as the number and importance of the evaluation indicators selected. Therefore, it is necessary to conduct more in-depth exploration and research on the risk assessment of highway bridge construction, in order to move forward in the direction of sustainable development.

**Author Contributions:** Conceptualization, J.Z. and J.F.; Data curation, Q.L. and J.F.; Investigation, Q.L.; Methodology, J.Z.; Resources, Q.L. and J.Z.; Software, Q.L. and J.F.; Supervision, J.Z.; Validation, Q.L.; Visualization, J.F.; Writing—original draft, Q.L. and J.F.; Writing—review and editing, J.Z. All authors have read and agreed to the published version of the manuscript.

**Funding:** This research received no external funding.

**Institutional Review Board Statement:** Not applicable.

**Informed Consent Statement:** Not applicable.

**Conflicts of Interest:** The authors declare no conflict of interest.

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
