# Peer review of "Safety Risk Assessment of Highway Bridge Construction Based on Cloud Entropy Power Method"

_applsci, doi:10.3390/app12178692_

Round 1
Reviewer 1 Report
The manuscript “Safety Risk Assessment of Highway Bridge Construction Based on Cloud Entropy Power Method”, noted as infrastructures-1875863, developed an improved approach to better weight different factors and reflect the uncertainty during bridge construction. The Cloud method and Cloud images were used to quantitatively categorize the risk level. And the improved method was validated with consistency by the AHP method.
Overall, the manuscript was clearly written. Methods were well explained and equations were sufficiently provides. However, this manuscript still needs to be revised and improved in the following aspects:
------------------------------------------------------------
1. The authors mentioned that 4 primary and 12 secondary indicators were selected to study in this manuscript but at the same time there are other factors would also affect the bridge construction risk. The authors should list some of the other factors which could provide readers a complete picture. Also the authors should give their reasons why these factors were not considered.
------------------------------------------------------------
2. Line 293: the authors proposed using Equation 13 from reference [31] to weight difference factors. It seemed Reference [31] already applied this equation in the risk quantification. Could the authors elaborate more on the difference between their novelty and reference [31]?
------------------------------------------------------------
3. Figure 7: Please add legends to make readers better understand the blue and red dots.
------------------------------------------------------------
4. The comparative study between the new Cloud method and the AHP showed that the two approaches reach similar risk evaluation with consistency. The benefits of the new Cloud method seemed to be undermined by the comparative study. Stronger arguments or proofs will be helpful to demonstrate the advantage of the new Cloud method in the comparative study.
------------------------------------------------------------
5. This manuscript needs to carefully refined as there’re grammatically errors, repetition of sentences and paragraph. The English writing should also be improved to avoid words like “in my country (Line 33)”.
------------------------------------------------------------
6. Reference styles need to be corrected and unified. And several papers were referred for the authors’ consideration of using machine learning in risk quantification and handling multiple factors.
Yuan, X., Chen, G., Jiao, P., Li, L., Han, J. and Zhang, H., 2022. A neural network-based multivariate seismic classifier for simultaneous post-earthquake fragility estimation and damage classification. Engineering Structures, 255, p.113918.
Reviewer 2 Report
Overall this is quite a good paper but suffers from some significant detractions. In particular, the first two sentences get the paper off to a bad start as they appear and read like Chinese propaganda. As those first two sentences add nothing to the paper they should be dropped. Sentence 2 uses the first-person word 'my' which in any case is inappropriate [should use third person passive voice]. Note there are 3 authors. Also drop ref [1], as this is not accessible outside China. Instead an acknowledgement section could be added at the end descibing the contribution of the each author. Risk accident and safety accident seem awkward expressions, especially the latter as it is strictly an oxymoron. Please modify accordingly.
Table 1, change middle to medium.
The conclusions are too long winded, they could be tightened up and made short. Suggest more pithy conclusions points preceeded by a narative-based discussion of key points/finding/observations.
Although the English, overall, is not bad, it could be improved upon. I strongly suggest it is re-editied by a NATIVE speaker of the English language [means not an expert in your country, as to me there would remain a wide gulf in terms of final readability].
